# Loss of surface transport is a main cellular pathomechanism of CRB2 variants causing podocytopathies

Annika Möller-Kerutt[1],* ⬛, Birgit Schönhoff[1],*, Yvonne Rellmann[2], Britta George[1], Daniela Anne Braun[1], Hermann Pavenstädt[1], Thomas Weide[1] ⬛

**Crumbs2 (CRB2) is a central component of the renal filtration barrier and part of the slit diaphragm, a unique cell contact formed by glomerular podocytes. Some *CRB2* variants cause recessive inherited forms of steroid-resistant nephrotic syndrome. However, the disease-causing potential of numerous *CRB2* variants remains unknown. Here, we report the establishment of a live-cell imaging–based assay, allowing a quantitative evaluation of the pathogenic potential of so far non-categorized *CRB2* variants. Based on in silico data analysis and protein prediction software, putative disease–associated CRB2 missense variants were selected, expressed as CRB2-GFP fusion proteins, and analyzed in reporter cell lines with BFP-labeled plasma membrane. We found that in comparison with PM-localized WT, disease-associated CRB2 variants remained predominantly at the ER. Accumulation at the ER was also present for several non-characterized CRB2 variants and variants in which putative disulfide bridge–forming cysteines were replaced. Strikingly, WT CRB2 retained inside the ER in cells lacking protein disulfide isomerase A3, indicating that posttranslational modification, especially the formation of disulfide bridges, is a crucial step for the CRB2 PM transport.**

## Introduction

In mammalian kidneys, podocytes are essential cells of the glomerular filtration barrier (GFB). Between their branched cellular protrusions, called foot processes (FPs), these postmitotic cells form a unique junction, called a slit diaphragm (SD). Together with the glomerular basement membrane and the fenestrated endothelium, the SD establishes the blood–urine barrier (Pavenstädt et al, 2003; Grahammer et al, 2013; Scott & Quaggin, 2015). On molecular level, the SDs' tasks are mediated by multiprotein complexes containing central single-pass membrane proteins that bridge the distance between neighboring FPs.

Injury of the SD causes loss of serum proteins in the urine. This proteinuria is a hallmark of many glomerular diseases and precedes renal failure (Wiggins, 2007; Kopp et al, 2020). Remarkably, many proteinuric renal diseases appear in childhood and are monogenetic inherited diseases caused by mutations in podocyte-specific genes (Sadowski et al, 2015; Vivante & Hildebrandt, 2016; Kopp et al, 2020). The best-studied SD protein is Nephrin (gene *NPHS1*), and more than 250 Nephrin mutations have been linked to congenital nephrotic syndrome of the Finnish type (CNF), an inherited form of steroid-resistant nephrotic syndrome (SRNS) leading to severe proteinuria and renal failure (Kestilä et al, 1998; Martin & Jones, 2018).

In 2015, two studies also identified mutations in the human *CRB2* gene associated with a phenotype resembling the phenotype caused by Nephrin mutations (Ebarasi et al, 2015; Slavotinek et al, 2015). Taking advantage of mice lacking CRB2 exclusively in podocytes, CRB2 was identified as an essential component of the glomerular filtration barrier (Möller-Kerutt et al, 2021; Tanoue et al, 2021). The phenotype of these mice strongly resembled the one of patients with mutations in the *CRB2* gene and has also striking clinical similarities to SRNS forms caused by mutations in single-pass membrane proteins of the SD, such as *FAT1*, *KIRREL*, and *NPHS1* (Kestilä et al, 1998; Gee et al, 2016; Martin & Jones, 2018; Solanki et al, 2019).

Podocyte cell lines expressing GFP-tagged CRB2 fusion proteins carrying the fluorescent protein tag inside the extracellular domain (ECD) showed the same cellular CRB2 as cells expressing untagged CRB2 (Möller-Kerutt et al, 2021). Based on these results, we now established a novel and improved cell-based assay to investigate the pathogenic potential of CRB2 variants including also far uncharacterized CRB2 allelic variants. Using this in vitro system in combination with quantitative live-cell imaging, we discovered a reduced transport from the ER to the plasma membrane (PM) as a common cellular pathomechanism for both well-known renal

---

[1]University Hospital of Muenster (UKM), Internal Medicine (MedD), Muenster, Germany   [2]Institute of Physiological Chemistry and Pathobiochemistry, University of Muenster, Muenster, Germany

Correspondence: weidet@uni-muenster.de; a_moel20@uni-muenster.de
*Annika Möller-Kerutt and Birgit Schönhoff contributed equally to this work

failure–causing CRB2 variants and CRB2 variants with a predicted but so far non-validated pathogenic potential. Moreover, we elucidated differences between various SRNS-causing CRB2 variants concerning their ER-to-PM transport efficiency. Finally, our results identified disulfide bridge formation as a crucial precondition for ER-to-PM transport of CRB2.

## Results

### Identification of CRB2 variants with a putative pathological potential

Only a few CRB2 variants are well documented and characterized yet, and much less is known about the pathogenic potential of additional variants within the human *CRB2* gene (Ebarasi et al, 2015; Slavotinek et al, 2015; Lamont et al, 2016). To address this, we used publicly available databases ClinVar and gnomAD and started with an in silico analysis to obtain an overview of reported CRB2 variants (Lek et al, 2016; Landrum et al, 2018; Karczewski et al, 2020). Our approach focused on missense mutations and excluded synonymous mutations as they do not change the aa sequence. We also excluded mutations within splice regions, and nonsense (*stop-gained*) and frameshift mutations as they usually result in truncated versions of the CRB2 protein (leading to *loss-of-function* phenotypes).

First, we selected the more than 150 CRB2 missense variants listed in ClinVar because of their correlations to human health. For further experimental analyses, 16 CRB2 variants of them were chosen, 14 either with a putative disease–associated classification (in ClinVar designated as *likely pathogenic*, *pathogenic*, or *uncertain significance*) or with a link to a renal phenotype, and two putative *benign* variants (M145T and R610W), which show also high allele frequencies in the gnomAD (Table S1). These CRB2 mutations or variants, their mode of inheritance, and available information about clinical features are summarized in Table 1.

### Establishment of a live-cell imaging test system to evaluate the pathological potential of CRB2 missense variants

Previously, we used cell lines with a podocyte background to analyze the intracellular localization of GFP-tagged CRB2 WT reference and four well-documented disease-associated variants in the ECD (C629S, R633W, N800K, and R1249Q). This study displayed that in contrast to CRB2 wt, missense variants remained in the ER and showed a *loss-of-surface-transport* phenotype (Möller-Kerutt et al, 2021). AB8/13 are large flat cells and well suited for immunofluorescent staining and live-cell imaging. However, their handling is time-consuming and the cells are difficult to transfect. To overcome these technical issues, we generated a stable HEK293T cell line, which constitutively expresses blue fluorescent protein (BFP) fused to a prenylation motif (CAAX), which targets the BFP-CAAX to the PM. This system allows a robust live-cell imaging readout for estimation of GFP-tagged CRB2 at the PM (Fig 1A). To validate the system, we tested first the CRB2-GFP reference (wt) and second the well-characterized disease-causing variants, which we previously

analyzed in cultured podocytes (Möller-Kerutt et al, 2021). In addition to the PM, we also labeled the ER with ER-Tracker. In this HEK293T-based in vitro system, the signal of CRB2-GFP wt and the R610W variant (classified as *likely benign*) strongly merged with the BFP-labeled PM, whereas disease-associated CRB2 variants C629S, R633W, N800K, and R1249Q mainly colocalized with the ER (Fig S1). Thus, the cells showed a similar intracellular distribution of the CRB2-GFP signal compared with the stable immortalized podocytes (Möller-Kerutt et al, 2021).

In the next step, non-characterized CRB2 missense variants were analyzed (Table 1 and Fig 1B). We observed three patterns: first, a pattern like the reference protein (wt) in which most of the cells show CRB2-GFP at the blue-labeled PM, like M145T or R610W (Figs 1C and S1). The second pattern exhibits an intracellular CRB2 distribution that mainly merges with the ER labeling, like C620S (Fig 1C). This pattern was observed in most of the analyzed CRB2 missense variants (C384F, R534W, R628C, E634A, P1064S, G1088D, T1187P, or G1205S; Fig S2). Finally, we found an intermediate distribution for variants R1072C and R1249Q (Figs 1C and S1), showing CRB2 pools at the PM and the ER. Next, we selected the variants M145T, C620S, and R1072C (one for each observed localization phenotype) for confirmation in immortalized podocytes. These CRB2 variants showed similar results as observed in HEK293T cells (Fig S3).

For evaluation of the live-cell imaging data, we applied a *PM localization* score and determined the ratio between cells with PM localization (total or partial) versus the total number of CRB2-GFP–expressing cells (including cells with no PM localization). Quantification of the data (>300 cells per approach and variant) showed a strong ER restriction for most of the disease-associated CRB2 missense variants. Here, less than 10% of the cells showed a signal at the BFP-labeled PM (Fig 1D and Table S2). Of note, missense variants R1072C and R1249Q reached *PM localization* of 32.2% and 20.7%, respectively, which is in line with the intermediate distribution patterns shown in Figs 1C and S2. In contrast, the amount of PM localization of M145T and R610W was similar to that of the CRB2 reference with a score above 80% (Fig 1D).

In addition to this cell population's score, the HEK293T-BFP-CAAX in vitro system also allows the determination of the CRB2-GFP signal overlay with BFP at the PM per cell. This *PM localization per cell* score is based on a Pearson correlation coefficient r, with values between "+1" (perfect correlation with the PM) and "−1" (perfect anti-colocalization). The data based on this evaluation are given in Fig 1E. Interestingly, the variant R610W, classified as *benign* or *likely benign*, slightly differs from variant M145T and the CRB2 reference (wt), demonstrating variability even in "WT–like" variants that mainly localize at the PM. However, the mean r-value was positively correlated, demonstrating that in most analyzed cells, this variant reaches the PM.

For all disease-associated CRB2 variants, the r-values were negatively correlated. Thus, there is no significant colocalization between the PM-BFP and the CRB2-GFP signal. Exceptions are CRB2 variants R1072C and R1249Q. Here, several individual cells reach positive signal r-values, suggesting that these variants might be less severely affected than the other ones (Fig 1E). Together, the observed *loss-of-surface-transport* phenotype of *CRB2* missense variants indicates that only small amounts of mutated CRB2 proteins reach the SD target region in vivo.

**Table 1.   CRB2 variant used in this study: the table summarizes the information about 14 CRB2 variants (out of over 150 missense annotations) that are categorized with possible pathological relevance in ClinVar.**

| Variant | Clinical significance/ phenotype condition | Homozygous versus compound heterozygous mutation | Age of onset (gender of patient) | Reference | ClinVar accession |
|---|---|---|---|---|---|
| M145T | Benign | — | | — | VCV001166113.1 |
| C384F | Uncertain significance, FSGS9 | — | | — | VCV000830021.1 |
| R534W | Uncertain significance, FSGS9 | — | | — | VCV000830021.1 |
| R610W | (likely) Benign | — | | — | VCV000773229.4 |
| C620S | Pathogenic | Homozygous | 4 yr | Ebarasi et al (2015) | VCV000180699.2 |
| | VMCKD, FSGS9 | | 6 yr | | |
| R628C | (likely) Pathogenic | Heterozygous (with G1036AfsTer43 or G839W) | 9 mo | Ebarasi et al (2015) | VCV000180700.2 |
| | SRNS, FSGS9 | | 3 yr (female) | Watanabe et al (2018) | |
| C629S | Pathogenic SRNS, FSGS9 | Homozygous | 3 yr | Ebarasi et al (2015) | VCV000180702.1 |
| R633W | Pathogenic VMCKD | Homozygous | 20th gestation wk | Slavotinek et al (2015) | VCV000180707.1 |
| | | | 7 mo | | |
| E643A | (likely) Pathogenic/uncertain significance | Heterozygous (with N800K) | 17th gestation wk (female) | Slavotinek et al (2015) | VCV000180706.2 |
| | VMCKD | | 16th gestation wk (male) | Lamont et al (2016) | |
| N800K | (likely) Pathogenic/ uncertain significance | Heterozygous (with W759Ter or G1036AfsTer43) | 18th gestation wk (female and male) | Slavotinek et al (2015) | VCV000546072.5 |
| | | | | Date et al (2019) | |
| | VMCKD | | | Jaron et al (2016) | |
| P1064S | Uncertain significance | Heterozygous with T902M or E149Ter | 7 yr (male) | Fan et al (2018) | VCV000522525.1 |
| | SRNS, FSGS9 | | | | |
| R1072C | Likely pathogenic | Heterozygous with C609Ter | 4 yr (male) | Domingo-Gallego et al (2022) | VCV000974518.1 |
| | SRNS, FSGS9 | | | | |
| G1088D | Likely pathogenic | — | | — | VCV000430364.2 |
| T1187P | Uncertain significance | — | | — | VCV000522526.1 |
| | FSGS9 | | | | |
| G1205S | Uncertain significance | Heterozygous with R764Ter | 11 yr (female) | Lu et al (2021) | VCV000829899.1 |
| | SRNS, FSGS9 | | 8 yr (male) | | |
| R1249Q | Uncertain significance (pathogenic and benign) | Homozygous | | Ebarasi et al (2015) | VCV000180703.4 |
| | SRNS, FSGS9 | | | | |

For some of them, patient studies with phenotype anamneses exist. In addition, two benign missense variants (M145T and R610W) are listed, which are used as WT–like controls in this study. VMCKD, ventriculomegaly with cystic kidney disease.

## Cysteine variants in the 10th EGF-like repeat change disulfide bridge formation in CRB2

The ECD of CRB2 is composed of three laminin G (LG) domains and 15 EGF-like repeats. Looking at the distribution of the selected CRB2 missense variants found in ClinVar, we identified several missense variants affecting cysteine residues and recognized an accumulation of putative pathological missense variants within the 10th EGF repeat (Fig 1B and Table S3).

EGF-like repeats are evolutionarily conserved protein–protein-interacting modules initially identified for the EGF (Wouters et al, 2005). Each EGF-like repeat has six cysteine residues that form intradomain disulfide bridges, which are crucial for folding of this domain (Wouters et al, 2005). In case of the 10th EGF repeat, these relevant disulfide bridges are formed by the cysteine pairs, C609 ↔ C620, C614 ↔ C629, and C631 ↔ C640. Interestingly, for all these cysteine pairs missense variants exist, including C620S and C629S that have been linked to SRNS before (Ebarasi et al, 2015; Slavotinek et al, 2015). For positions C614 and C631, two different missense varinats (C614S, C614Y and C631F, C631F) are listed in the genome data set of gnomAD. In addition, there are allelic *CRB2* variants in which aa changes lead to additional cysteines, like R605C, F627C, and the pathogenic R628C (Ebarasi et al, 2015). These variants might form alternative disulfide bond pair variants, differing from the

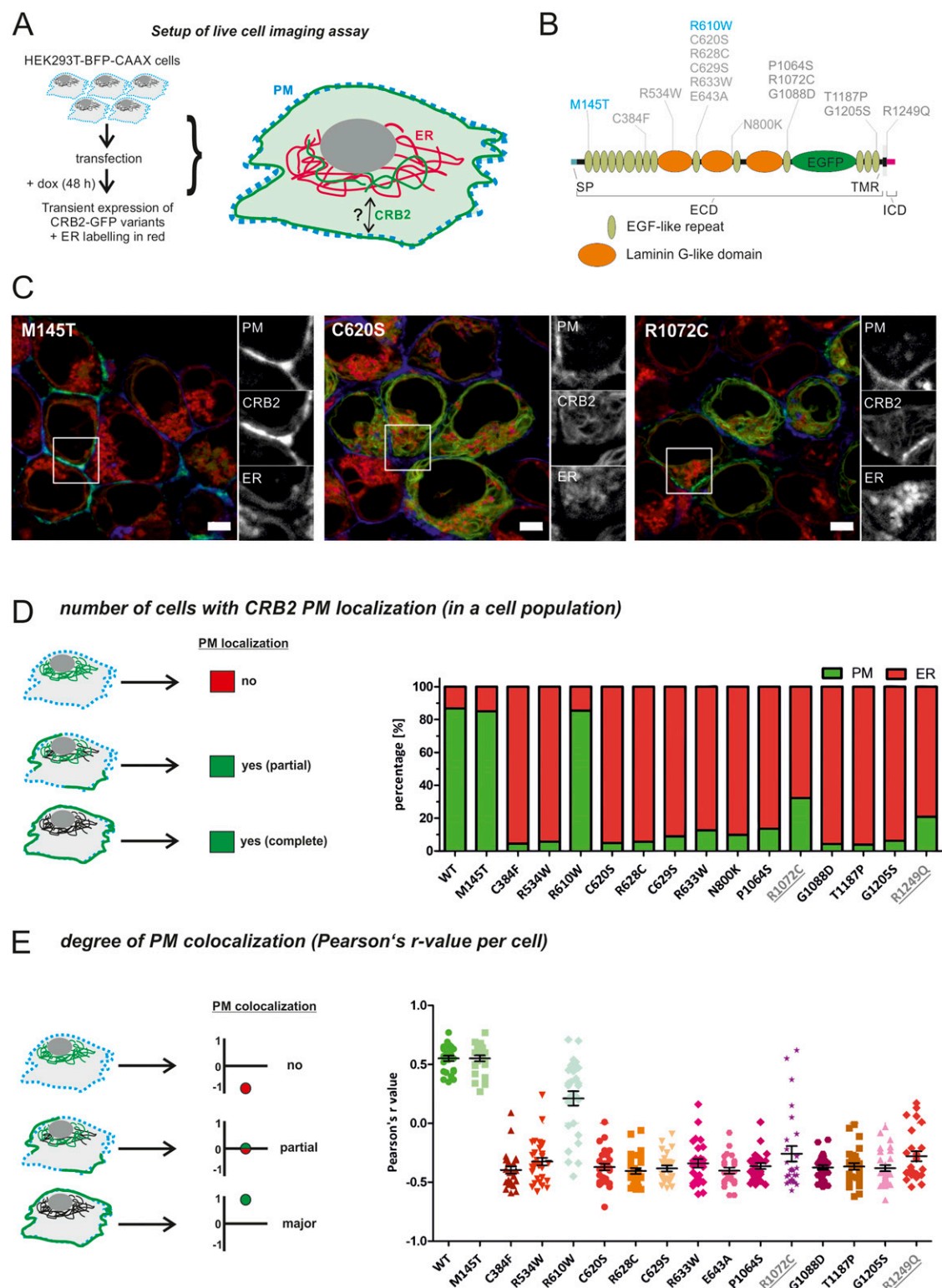

**Figure 1. Disease-associated missense variants lead to retention of CRB2 in the ER.**
**(A)** Setup: cells constitutively expressing BFP-CAAX to label the PM were used to investigate the localization GFP-tagged CRB2 wt or missense variants. **(B)** Scheme of CRB2-GFP with marked disease-associated variants (gray) and potential benign control variants (blue). SP, signal peptide; TMR, transmembrane region; ECD, extracellular domain; ICD, intracellular domain; ER, endoplasmic reticulum; PM, plasma membrane. **(C)** Images of three CRB2 variants used in live-cell assay. Gray-scale details of the PM = BFP-CAAX signal, CRB2 = GFP, and ER = red are shown on the right of each merged image. Scale bar = 5 $\mu$m. **(D)** Quantification of cell number with CRB2-GFP PM localization. Cells with a GFP signal at the PM were grouped to "PM" (scheme at the left). In most cells overexpressing missense CRB2 variants, the GFP-CRB2 signal is mainly at the ER. The detailed percentages of all analyzed CRB2 variants are summarized in Table S2. **(E)** Degree of PM colocalization per cell: the r-values of the disease-associated CRB2 variants are negative and show no colocalization with the PM (values around −0.5). **(D, E)** Welch's *t* test for disease-associated variants versus wt: *P* < 0.001 (D, E). ER, endoplasmic reticulum; PM, plasma membrane; N = 3.

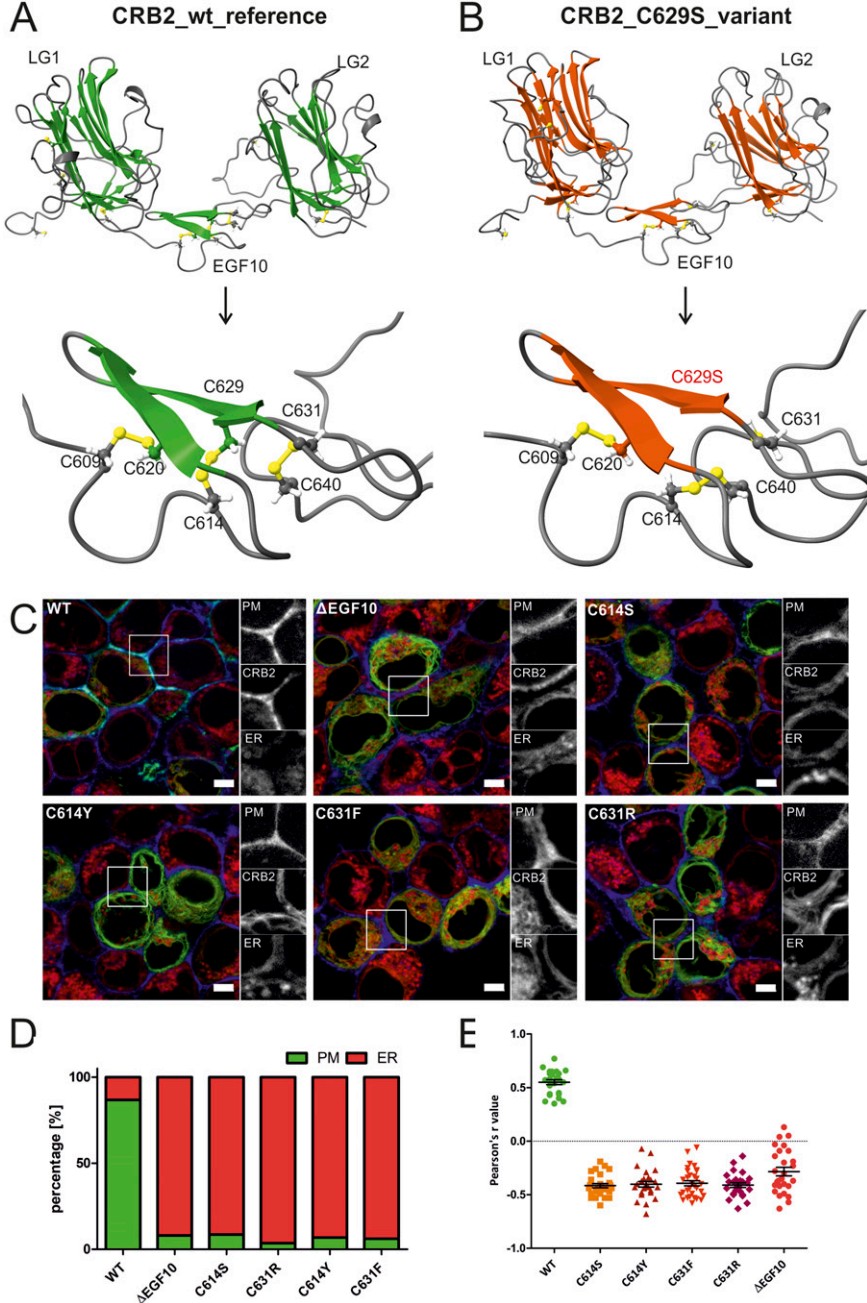

**Figure 2. Cysteine-affected variants in the EGF10 of CRB2 display an altered disulfide formation and ER retention phenotype.**

**(A, B)** 3D model of LG1, EGF10, and LG2 domains (region aa 431–808) from CRB2 wt (A) and disease-associated CRB2 variant C629S (B) calculated with *RoseTTAFold* and visualized in ChimeraX. The lower panel shows disulfide bonds (labeled in yellow) between the different cysteine residues within the EGF10. In case of the C629S variant, only two of three disulfide bridges of EGF10 are formed. **(C)** Live-cell images of the GFP-tagged CRB2 reference (wt), CRB2 without EGF10 (ΔEGF10), and cysteine missense variants at positions 614 and 631, respectively. Gray-scale details for the signals of BFP-CAAX (PM), GFP (GFP-tagged CRB2), and ER-Tracker red (ER). Scale bar = 5 µm. **(D)** *PM localization* score: quantification of ≥300 cells for GFP localization within the cell. In contrast to the CRB2 reference (>86% PM localization), CRB2 variants show strongly reduced pools at the PM (<10%). **(E)** *PM localization per cell* score: colocalization of the blue and green pixels was determined with Fiji-Coloc2 in n = 25 cells. The colocalization r-value for the CRB2 reference protein (wt) shows a positive correlation. In cells expressing GFP-CRB without EGF10 (ΔEGF10), the correlation is negative, similar as for analyzed variants with replaced cysteines. Welch's *t* test for variants versus wt: *P* < 0.001. Abbreviations: ER, endoplasmic reticulum; PM, plasma membrane; N = 3.

predicted ones for the 10th EGF. Except for the disease-associated C620S, C629S, and R628C variants, the putative pathogenic potential has not been tested so far.

For a closer look into these cysteines relevant for disulfide bridges, we applied the *RoseTTAFold* protein folding software tool, which uses deep learning to predict 3D protein structures based on the aa sequence (Yang et al, 2020; Baek et al, 2021). As region of interest, we selected the 10th EGF-like repeat (aa 605–640) with the flanking LG1 and LG2 domains (CRB2 aa 431–808). The 3D images with cysteine disulfide bridges (given in yellow) were visualized by *ChimeraX* (Pettersen et al, 2021). Fig 2A and B shows example images

of the in silico studies for CRB2 wt and the C629S variant. In case of the C629S variant, the C614 ↔ C629 disulfide bridge is missing and a novel one formed between C614 and C640 is predicted in *ChimeraX*, replacing the disulfide bonds C614 ↔ C629 or C631 ↔ C640, respectively. The structure between LG1 and LG2 seems to be preserved, and the disulfide bridges C579 ↔ C 603 and C766 ↔ C805 of the flanking LG1 and LG2. In case of C614S and C614Y variants, the 3D software indicates an additional loss of the neighboring disulfide bridge C631 ↔ C640, accompanied by de novo disulfide formation between C629 and C640 for C614S/C614Y variants and between C614 and C640 for the pathogenic C629S

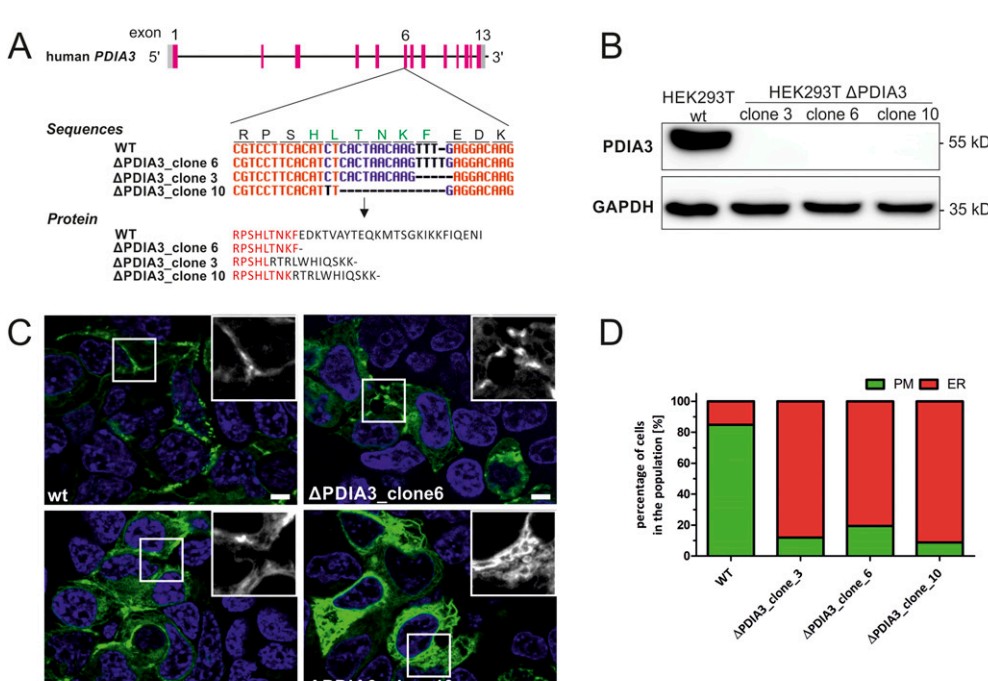

**Figure 3.   PDIA3 KO results in defective processing of CRB2 wt.**
**(A)** Schemes of the human *PDIA3* gene (upper image), the DNA modifications that caused CRISPR/Cas9-mediated changes (middle image), and the amino acids of the truncated PDIA3 proteins (lower image). **(B)** Western blot analysis of HEK293T ΔPDIA3 KO cell lines to validate the absence of PDIA3 on protein level. **(C)** Localization of the CRB2-GFP WT reference in standard HEK293T cells and HEK293T cells lacking ΔPDIA3 (clone nos. 3, 6, and 10). Scale bar: 5 *µ*m; nuclei = Hoechst stain in blue. **(D)** Quantification of cell number with CRB2-GFP PM localization. In control cells, CRB2-GFP mainly localizes at the PM (~84%). In HEK293T cells, PM pools of GFP-CRB2 WT are strongly reduced (≤20%); N = 3.

variant. The results of the in silico 3D prediction studies are summarized in Table S3.

Based on these 3D predictions, we included so far non-characterized C614S, C614Y, C631R, and C631F allelic variants of CRB2 in our live-cell imaging studies (Fig 2C) and used both scores (*PM localization* and *PM localization per cell*) to quantify the results (Fig 2D and E). We also designed an artificial deletion mutant lacking the complete 10th EGF repeat (aa 605–640) that includes several variants with known or anticipated pathological potential (Fig 1B). As expected, these CRB2 variants and the ΔEGF10 deletion mutant significantly failed to reach the PM and showed ER retention in almost all counted cells (Fig 2C–E). In comparison with the reference, all variants also showed negative colocalization r-values (about –0.5), indicating that only small CRB2-GFP fractions reached the PM. Thus, these so far non-characterized variants may have an increased pathogenic potential.

### Disulfide bridge formation as a crucial factor for CRB2 transport to the cell surface

Protein folding of SD transmembrane proteins is a complex process, which requires chaperon-assisted folding of subdomains. Thus, we hypothesized that CRB2 is a possible target for the protein disulfide isomerase A3, PDIA3 (also known as ERp57). PDIA3 localizes to the ER lumen and interacts with the chaperones calreticulin and calnexin to modulate folding of newly synthesized glycoproteins (Kanemura et al, 2020; Powell & Foster, 2021).

We generated HEK293T *PDIA3 KO* cells and selected three independent cell clones for further analyses (Fig 3A and B). In cells lacking PDIA3, most of the CRB2-GFP wt signal was retained in the ER, and only 10–20% reached the PM (Fig 3C). In the control cell line, more than 80% of the cells showed predominant PM localization of

CRB2-GFP wt (Figs 1 and 2). Thus, *KO* of *PDIA3* results in a loss of CRB2 transport to the PM, similar as demonstrated for disease-associated CRB2 variants or CRB2 variants with predicted increased high pathogenic potential (Table S1 and Figs 1, 2, S1, and S2). Taken together, these data suggest that PDIA3 activity is a central cofactor for proper processing and subsequent transport of CRB2 to its target region.

## Discussion

In this study, we report the development of a robust in vitro assay, suitable to confirm or validate given CRB2 disease-associated variants and as a *screening tool* to investigate the pathogenic potential of novel, so far uncharacterized CRB2 variants. Our data revealed that CRB2 variants with a predicted or putative pathogenic potential showed similar severe transport defects as the validated disease-associated CRB2 variants with clear genotype–phenotype correlation (Ebarasi et al, 2015; Slavotinek et al, 2015; Möller-Kerutt et al, 2021). This suggests (i) that CRB2 accumulation inside the ER is a common feature of CRB2 disease-associated variants and (ii) that proteinuric phenotypes caused by CRB2 are in principle *loss of CRB2-SD–targeting* phenotypes. This could be of particular relevance also for other proteins, as SD proteins such as CRB2 have to cover long distances from the perinuclear region (ER) toward highly branched FPs in the cell periphery to be finally inserted into the SD.

Thus, disrupted SD protein trafficking could be a novel and underestimated pathomechanism for proteinuric renal diseases in general. Indeed, this assumption is supported by data from Liu and colleagues, who identified transport defects for the SD protein Nephrin as causative for SRNS, and by more recent data from Ebarasi et al, showing a lack of *crb2b* in zebrafish leads to trafficking

defects of its binding partner Nephrin (Liu et al, 2001, 2004; Nishibori et al, 2004; Ebarasi et al, 2009; Möller-Kerutt et al, 2021).

The PM localization scores used in our studies revealed only ~10–20% of cases where CRB2 variants with pathogenic potential or variants already linked to CRB2-related syndrome are localized at the PM. Thus, ER-to-PM transport is strongly reduced but not completely inhibited for disease-associated CRB2 variants. Vice versa, even the WT reference version of CRB2 is not completely exported to the cell surface, indicating that the ER-to-PM transport of SD proteins is a highly dynamic process and renal failure linked to the *CRB2* gene is most probably caused by podocytes that fail to transport CRB2 to the SD. Thus, CRB2 ER-to-PM transport has to ensure a certain threshold level at the PM (or in vivo at the SD).

Protein transport requires glycosylation, disulfide bridge formation, and chaperone-assisted 3D folding. Indeed, the impressive relevance of glycosylation for proper export of SD proteins has been recently demonstrated by mice lacking cytidine monophosphate N-acetylneuraminic acid synthetase (*Cmas* gene) in podocytes. The study identified N-glycan capping with sialic acid as an essential precondition for the localization in the SD proteins (Niculovic et al, 2019).

Our data now show that also the formation of disulfide bridges is a prerequisite for proper SD protein trafficking, as cells lacking PDIA3 showed a similar defective ER to PM transport for WT CRB2, CRB2 variants with pathogenic potential, CRB2 disease-associated missense variants, or the artificial CRB2 deletion mutant without the central 10th EGF-like repeat (ΔEGF10).

Protein disulfide isomerase activity inside the ER lumen seemed to be an essential factor for correct formation of up to 48 functional disulfide bridges within the CRB2 ECD (44 within the EGF-like repeats and four in LG domains). Thus, dysfunctional disulfide isomerase activity could be a further relevant causative for podocytopathies. How disturbances and delays of the SD processing and posttranslational modifications contribute to human inherited forms of SRNS podocytopathies requires further analyses. The here described assay system is perhaps a suitable technical tool for testing of the pathological potential of novel uncharacterized CRB2 protein variants. In addition, it might also serve as a cell-based platform for evaluating the therapeutic potential of agents (including chemical chaperones) influencing CRB2 folding and export to the cell surface or the SD.

# Materials and Methods

### Database search for CRB2 variants

To obtain information about documented CRB2 variants, we used data from ClinVar (Clinically relevant Variation), gnomAD (Genome Aggregation Database), and UniProtKB (Universal Protein KnowledgeBase) databases (Landrum et al, 2018; UniProt Consortium, 2019; Karczewski et al, 2020). For the putative clinically relevant CRB2 variants, we predominantly took advantage of the ClinVar database that includes preliminary pathological classifications or interpretations (Landrum et al, 2018). Information about *CRB2* allele frequencies was received from the gnomAD based on exome and whole-genome sequences compared with GRch37/hg19 reference (Karczewski et al, 2020).

### Constructs and cloning

Cloning and site-directed mutagenesis of CRB2 variants are based on a pENTR plasmid carrying a cDNA insert encoding for fusion protein of CRB2 wt reference (aa 1–1285) and a EGFP within the 13th EGF-like repeat of the CRB2 ECD (inserted after an aspartate at position 1095) as reported previously (Möller-Kerutt et al, 2021). The CRB2 reference (WT) is according to UniProt sequence Q5IJ48 with the exceptions of common natural allelic variants at positions 90 (asparagine instead of threonine, T90N), 709 (V709A), and 969 (T969A). A pENTR-CRB2-GFP plasmid was also used to generate the CRB2 mutant lacking the 10th EGF-like repeat (ΔEGF10, missing aa 605–640). All pENTR plasmids with mutagenized *CRB2-GFP* cDNA inserts were validated via sequencing and shuttled into pINDU-CER21_Puro plasmids using Gateway LR Clonase (Thermo Fisher Scientific) according to the manufacturer's instructions and as described earlier (Schulze et al, 2014; Granado et al, 2017). Gateway shuttling was also used to transfer the insert of the pME-mTagBFP-CAAX (Don et al, 2017) construct (a kind gift from Nicholas Cole, Addgene no. 75149) into a modified pQCXIP-GW vector. All details concerning constructs and primers are summarized in Table S4.

### Cell culture and generation of stable cell lines

HEK293T and AB8 immortalized podocytes were cultivated as described earlier (Schulze et al, 2014; Granado et al, 2017) and transfected using Lipofectamine 2000 (Thermo Fisher Scientific) according to the manufacturer's instruction. The HEK293T-BFP-CAAX reporter cell line and stable AB8 cell lines for CRB2 variants were generated as previously described (Schulze et al, 2014; Granado et al, 2017; Möller-Kerutt et al, 2021). Briefly, retroviral particle production was performed in HEK293T cells transiently transfected with psPAX2 and pMD2.G helper plasmids and pQCXIP_BFP-CAAX plasmid. Supernatants containing virus particles pseudo-typed with VSV-G were collected and filtered through a 0.45-$\mu$m sterile filter (EMD Millipore). Fresh target HEK293T or AB8 cells were incubated with one volume of fresh DMEM/RPMI medium and one volume of the virus-containing filtrate supplemented with polybrene (final concentration 8 $\mu$g/ml). After 24 h, the virus particle–containing medium was replaced by fresh medium and cells were regenerated during the next 24 h followed by an additional transduction cycle of 24 h. Transduced cells were selected using puromycin (4 $\mu$g/ml for HEK293T, and 2 $\mu$g/ml for AB8).

### Establishing and validation of HEK293T *PDIA3 KO* cell lines

For generating a CRISPR/Cas9-based KO of the human *PDIA3* gene (also called ERp57) in HEK293T cells, we applied ERp57 CRISPR/Cas9 KO plasmids and ERp57 HDR plasmids (sc-401497) from Santa Cruz Biotechnology, as described before (Rellmann et al, 2019) and according to the manufacturer's instructions. In brief, HEK293T cells were cultivated in antibiotic-free DMEM (+10% FCS) and both plasmids were cotransfected with UltraCruz Transfection Reagent (sc-395739). After an incubation period of 48 h at 37°C, the medium was changed to normal DMEM (supplemented with 10% FCS and 1% PSG). Transfection efficiency was visually confirmed via fluorescent GFP (KO plasmid) and RFP (HDR plasmid) expression. Next, transfected

cells were selected with puromycin (4 µg/ml) for 1 wk. Afterward, single cells were seeded in 24-well plates to establish single clones.

Successful KO of the *PDIA3* gene was validated by sequencing of the targeted exon. For this, genomic DNA was extracted from the cells with lysis buffer (50 mM KCl, 1.5 mM MgCl2, 10 mM Tris–HCl, pH 8.3, 0.45% NP-40, and 0.45% Tween-20) containing 0.05 µg/µl Proteinase K (Qiagen) at 56°C overnight (ON). Samples were boiled for 10 min at 95°C before centrifugation for 10 min at 21,000*g*. The DNA-containing supernatant was used for PCR amplification of the respective *PDIA3* region of exon 6 before sequencing.

### Preparation of cell lysates

Cells cultured on cell culture plates were washed once with 1× PBS before lysis in an appropriate volume of RIPA buffer on ice (150 mM NaCl, 50 mM Tris, pH 8.0, 1% NP-40, 1% Triton X-100, 0.1% SDS, and 0.5% Na-deoxycholate, supplemented with complete protease tablets [Roche] and phosphatase inhibitor cocktails [Sigma-Aldrich]). Cells were scratched off the plate into a reaction tube, which was vortexed every 5 min within 30 min. After ultrasonication for 10 min, lysates were centrifuged for 15 min at 21,000*g* at 4°C. The supernatant was mixed with 2× Laemmli buffer before SDS–PAGE and immunoblotting.

### Western blotting

SDS–PAGE and Western blot analyses were performed as previously described (Granado et al, 2017; Weide et al, 2017). Briefly, cell lysates were boiled for 5 min at 95°C and equal volumes were separated via SDS–PAGE using 8% gels (Bio-Rad System). Afterward, proteins were transferred to a PVDF membrane (EMD Millipore) and incubated in blocking buffer containing 5% skim milk powder dissolved in TBS containing 0.05% Tween-20 (TBS-T) for 1 h at RT.

Primary antibodies against GAPDH (#14C10; Cell Signaling Technologies) and PDIA3 (HPA003230; Sigma-Aldrich) were diluted (1:1,000) in TBS-T with 5% BSA and incubated at 4°C ON. Next, the membrane was washed three times with TBS-T and incubated with horseradish peroxidase–coupled secondary antibodies (Jackson ImmunoResearch) diluted 1:3,000 in blocking buffer for 45 min at RT. After three further washing steps with TBS-T, the chemiluminescence signal was detected using a Clarity detection reagent (Bio-Rad) using an Azure Biosystems imager (c600; Bio-Rad).

### Live-cell imaging analyses of cells

HEK293T-BFP-CAAX cells or HEK293T ΔPDIA3 KO clones were seeded in Ibidi eight-well chambers (80826; Ibidi) and transiently transfected for 2 h with pIND21_CRB2-GFP plasmids using Lipofectamine 2000 according to the manufacturer's instruction (Thermo Fisher Scientific). Expression of CRB2 wt (reference sequence) variants was induced by 200 nM doxycycline (Dox) for 48 h. For labeling of the nuclei and the ER, cells were incubated with Hoechst 33342 (1 µg/ml; Invitrogen) and/or ER-Tracker red (1 µM; Invitrogen) for 30 min before quantitative live-cell imaging analyses of GFP-tagged CRB2. Cells were imaged in HBSS containing 30 mM Hepes using an Observer Z1 microscope equipped with Apotome 2.0, an AxioCam MRm camera (Zeiss), and Plan Apochromat 63×/1.40 Oil or EC Plan-

Neofluar 40×/1.30 Oil objectives. Images were processed with Fiji (http://fiji.sc/) or Zen software (Zeiss GmbH).

### Evaluation of live-cell images

We evaluated the colocalization of the CRB2-GFP signal with the PM marker BFP-CAAX signal as *PM localization score index*. In case of partial and complete CRB2-GFP localization with the PM marker, cells were evaluated as PM positive. Colocalization of CRB2-GFP with ER-Tracker red was judged as ER localization. For analyzed cell lines, at least 300 cells per cell line (≥15 independent images) were counted.

We also determined the degree of colocalization between the BFP-CAAX and CRB2-GFP at the PM per cell. To address that, the Coloc2 plugin of Fiji/ImageJ was used to measure the Pearson correlation coefficient (PSF 3.0; Costes' randomization = 100, Pearson's r-value [no threshold]). The BFP-CAAX signal (pixels) at the PM was used as a mask and compared with GFP-positive pixels. Pixel intensities of one channel were evaluated and compared with the intensity of the other channel and summarized in a scatterplot including the Pearson correlation coefficient. Thereby, "+1" indicated perfect correlation or colocalization, and "−1," perfect anti-colocalization. The Pearson correlation coefficient was ensured with the Costes significance test ($P > 0.8$) (Gavrilovic & Wählby, 2009). For both quantifications, only CRB2-overexpressing cells facing other CRB2-expressing cells were considered.

### Statistics

Tests for statistical significance of normally distributed data were performed using GraphPad Prism software with an unpaired two-tailed Welch *t* test for comparison between two groups of data. If not otherwise indicated, all data are given as the mean of at least three independent measurements (*$P < 0.05$; **$P < 0.01$; and ***$P < 0.001$).

### Protein prediction analysis

In silico protein 3D structure prediction was performed with *Robetta* as a protein structure prediction service that is continually evaluated through CAMEO and based on the *RoseTTAFold* modeling deep learning algorithm (Yang et al, 2020). Calculated 3D protein structures and prediction of disulfide bond formation were visualized with *ChimeraX* software (next-generation molecular visualization program from the Resource for Biocomputing, Visualization, and Informatics) (Pettersen et al, 2021).

## Supplementary Information

## Acknowledgements

We thank all members of the laboratory for critical discussions and Drs. Beate Vollenbröker and Rita Dreier for critical reading of the article. This work was supported by DFG grants to T Weide (WE 2550/2-2 and WE 2550/5-1)

and the Medizinerkolleg of Medical Faculty of the University of Münster (MedK 20-0056 to B Schönhoff). The work contains major parts of the MD thesis of B Schönhoff.

## Author Contributions

A Möller-Kerutt: conceptualization, data curation, formal analysis, supervision, validation, investigation, visualization, methodology, and writing—original draft, review, and editing.
B Schönhoff: conceptualization, data curation, formal analysis, validation, investigation, visualization, methodology, and writing—original draft.
Y Rellmann: resources, data curation, and formal analysis.
B George: data curation and formal analysis.
DA Braun: data curation and formal analysis.
H Pavenstädt: data curation, formal analysis, and funding acquisition.
T Weide: conceptualization, data curation, supervision, funding acquisition, project administration, and writing—original draft, review, and editing.

## Conflict of Interest Statement

The authors declare that they have no conflict of interest.

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
