## [Reviewer comments · Life Science Alliance]

Life Science Alliance

Loss-of-surface-transport is a main cellular pathomechanism of CRB2 variants causing podocytopathies

Thomas Weide, Annika Möller-Kerutt, Birgit Schönhoff, Yvonne Rellmann, Britta George, Daniela Braun, and Hermann Pavenstädt

DOI: <https://doi.org/10.26508/lsa.202201649>

Corresponding author(s): Thomas Weide, University Hospital Münster and Annika Möller-Kerutt, Uniklinik Münster, Medizinische Klinik D

Review Timeline:

Submission Date:	2022-08-04
Editorial Decision:	2022-09-09
Revision Received:	2022-12-08
Editorial Decision:	2022-12-08
Revision Received:	2022-12-12
Accepted:	2022-12-13

Transaction Report:

September 9, 2022

Re: Life Science Alliance manuscript #LSA-2022-01649-T

Prof. Thomas Weide
University Hospital Münster
Molecular Nephrology
Albert-Schweitzer-Campus 1, A14
Muenster D-48149
Germany

Dear Dr. Weide,

Thank you for submitting your manuscript entitled "Loss-of-surface-transport is a main cellular pathomechanism of CRB2 missense mutations causing podocytopathies" to Life Science Alliance. The manuscript was assessed by an expert reviewer, whose comments are appended to this letter. We invite you to submit a revised manuscript addressing the Reviewer comments.

When submitting the revision, please include a letter addressing the reviewer comments point by point.

Thank you for this interesting contribution to Life Science Alliance. We are looking forward to receiving your revised manuscript.

Sincerely,

B. MANUSCRIPT ORGANIZATION AND FORMATTING:

Reviewer #1 (Comments to the Authors (Required)):

1. The authors follow up on their interesting studies on the pathogenic potential of CRB2 variants for an intact slit diaphragm and consequences of disruption of CRB2 function for the glomerular filter. They now provide additional evidence for deleteriousness of CRB2 missense variants by demonstrating significantly impaired intracellular trafficking of both variants described in patients and yet undescribed variants. These findings are of relevance for better understanding the function of CRB2 for the podocyte/slit diaphragm function and the general disease relevance of impaired trafficking of SD proteins.

2. The presented data and the choice for conducted experiments is clear and concise. We do understand the choice of easy to handle and transfect HEK293 cells. However, it would strengthen the data to repeat a subset of the tested variants (for instance one benign, one intermediate and one deleterious variant) in immortalized podocytes by transient transfection of the variant and labeling of the PM.

3. There are multiple minor issues to be addressed:

- The authors should carefully reevaluate the use of the words 'variant' and 'mutation'
- page 3: should say: publicly available databases
- Fig 1 D: As authors have categorized into 3 groups, these should be visualized in the graph, potentially by displaying the partial group as yellow.
- page 5 bottom: should say: less severely affected
- page 6 top: Fig1A -> Fig1B
- page 7 first paragraph: revise truncated sentence that ends with 'includes several pathologically.'
- include in discussion if impaired post-translational modifications and/or trafficking could be ameliorated by drugs and if respective studies are planned for future studies.

Reviewer #1 (Comments to the Authors (Required)):

1. The authors follow up on their interesting studies on the pathogenic potential of CRB2 variants for an intact slit diaphragm and consequences of disruption of CRB2 function for the glomerular filter. They now provide additional evidence for deleteriousness of CRB2 missense variants by demonstrating significantly impaired intracellular trafficking of both variants described in patients and yet undescribed variants. These findings are of relevance for better understanding the function of CRB2 for the podocyte/slit diaphragm function and the general disease relevance of impaired trafficking of SD proteins.

Comment 1: We are very happy that the reviewer find our data interesting and of relevance for the field of podocyte /slit diaphragm function and disease in the context of CRB2.

2. The presented data and the choice for conducted experiments is clear and concise. We do understand the choice of easy to handle and transfect HEK293 cells. However, it would strengthen the data to repeat a subset of the tested variants (for instance one benign, one intermediate and one deleterious variant) in immortalized podocytes by transient transfection of the variant and labeling of the PM.

Comment 2: We thank the reviewer for that comment and agree, showing the phenotypes in an additional cell culture model preferably immortalized podocytes will strengthen the presented data.

However, there are some technical issues that should be considered: The mentioned HEK293 cell based assay system was established by a two-step process. In a first step we generated stable HEK293T cell lines expressing permanently the blue-fluorescent protein fused to the CAAX prenylation motif (BFP-CAAX). For that we used a retroviral-based system (RetroX) in combination with a pQCXIP expression vector that contains the BFP-CAX expression cassette. The transduced cells were selected by puromycin (because in addition to the gene of interest the pQCXIP vector backbone also mediates puromycin-resistance). In a second step we *transiently transfected* the HEK293T-BFP-CAAX cell lines with pINDUCER21_Puro plasmids. These plasmids encode for the different GFP-tagged CRB2 variants and induced its expression by the administration doxycycline. As HEK293 cells are easy to transfect with high transfection levels the system leads to robust and quantifiable results.

In principle we can follow a similar strategy for immortalized podocytes, but immortalized podocytes are very difficult to handle and show very low transfection rates, especially for transient transfections. (Indeed, to avoid this problem and establish a fast possibility to screen CRB2 variants was the main reason, why we established a HEK293T-based cell assays system.)

Nonetheless, we tried several approaches to transiently transfect immortalized podocytes by using different transfection agents, but all these approaches failed. Due to the very low transfection levels we obtained almost no cases in which two neighboring cells were positive for the expression of GFP-tagged CRB2 variants, which is essential for the evaluation of the CRB2 signal accumulating at the PM.

We also tried a sequential transduction strategy, by using first recombinant retrovirus (for BFP-CAAX) and next recombinant lentivirus particles (for GFP CRB2 expression, as described earlier by Möller-Kerutt et al., 2021). This approaches failed, as both recombinant viruses transfer a puromycin-resistance. In other words: clones selected by puromycin did not necessarily carry and express both genes of interest, BFP-CAAX and GFP-CRB2.

So we next tried a simultaneous transduction of podocytes with recombinant virus-mixtures, encoding for BFP-CAAX (retrovirus) and GFP-CRB2 fusion proteins (lentivirus). Here we obtained cell that expressed either BFP-CAAX, or GFP-labeled CRB2 or both (albeit in different ratios).

The images illustrate this problem in more detail.

Fig. 1: Simultaneous transduction of podocytes with recombinant virus-mixtures.

The parallel transduction of immortalized podocytes with recombinant virus mixtures to express BFP-CAAX (retrovirus) and GFP-CRB2 fusion proteins resulted in numerous cells that either BFP-CAAX (blue arrow) or CRB2-GFP. Only a sub-fraction minor of cells was positive for both (albeit in different ratios) (white arrows).

So finally, to address the reviewer's points and to confirm obtained results of the HEK293 system (in Fig. 1) we decide to follow the strategy of the foregoing study (Möller-Kerutt et al., 2021, JASN; Fig. 6 and suppl. Fig. S6). Here, we did the evaluation without a blue labelled plasma membrane and evaluated CRB2-GFP positive overlapping

regions between two cells and used and ER labeling as reference in human immortalized podocytes.

By using this setting, we analyzed the localization of the wildtype-like M145T CRB2 variant, the disease-associated C620S CRB2 variant and a variant with an intermediate cellular phenotype (R1072 CRB2 variant).

Quantification according to the protocol of *Möller-Kerutt et al* revealed that these CRB2 variants showed a same trend as observed in HEK293T cells. We included these data as novel suppl. Fig. to the manuscript.

3. There are multiple minor issues to be addressed:

-The authors should carefully reevaluate the use of the words 'variant' and 'mutation'

Comment 3: We thank the reviewer for this comment. Indeed, from geneticists it is sometimes recommended to avoid the word mutants, 1st, as it is not always clear at what point a variant becomes a (disease associated or –causing) mutant, and 2nd as even in case that variants are disease-associated it is not clear in how far their penetrance is also influenced by additional factors. Consequently, we replaced the word “mutants” by the “variants”, except for the artificial deletion mutants lacking the 10th EGF repeat (Δ EGF 10).

(Thus, variants that cause diseases are now called disease-associated variants, in cases where this is not clear we call them putative disease-associated variants.)

- page 3: should say: publicly available databases

Comment 4: We changed this term as recommended.

- **Fig 1 D:** As authors have categorized into 3 groups, these should be visualized in the graph, potentially by displaying the partial group as yellow.

Comment 5: In the Quantification in Fig. 1D we evaluated the CRB2-GFP between two GFP-positive cells, whether it is at the PM or not, independent of the degree of PM localization (whether it is full or partial). Differences in the degree of PM localization were assayed by co-localization analysis summarized in Fig. 1E.

To address reviewers point we underlined the CRB2 variants in the figure that represent the intermediate group.

- page 5 bottom: should say: less severely affected

Comment 6: We changed this term as recommended.

- page 6 top: Fig1A -> Fig1B

Comment 7: Thanks for this hint! We changed this as recommended.

- page 7 first paragraph: revise truncated sentence that ends with 'includes several pathologically.'

Comment 8: We now changed the truncated sentence into: “We also designed an artificial deletion mutant lacking the complete 10th EGF-repeat (aa 605-640) that includes several variants with known are anticipated pathological potential (Fig. 1B).”

- include in discussion if impaired post-translational modifications and/or trafficking could be ameliorated by drugs and if respective studies are planned for future studies.

Comment 9: We thank the reviewer for this comment: Indeed, *in vitro* system could be useful for high-throughput experiments, planned to the therapeutic potential of agents and drugs (including for example chemical chaperones). Therefore we added the passage:

“How disturbances and delays of the SD processing and posttranslational modifications contribute to human inherited forms of SRNS-podocytopathies requires further analyses. The here described assay system is perhaps a suitable technical tool for testing of the pathological potential of novel uncharacterized CRB2 protein variants. In addition, it might be also serve as a cell-based platform for evaluating the therapeutic potential of agents (including for example chemical chaperones) influencing CRB2 folding and export to the cell surface or the SD.”

December 8, 2022

RE: Life Science Alliance Manuscript #LSA-2022-01649-TR

Prof. Thomas Weide
University Hospital Münster
Molecular Nephrology
Albert-Schweitzer-Campus 1, A14
Muenster D-48149
Germany

Dear Dr. Weide,

Thank you for submitting your revised manuscript entitled "Loss-of-surface-transport is a main cellular pathomechanism of CRB2 variants causing podocytopathies". We would be happy to publish your paper in Life Science Alliance pending final revisions necessary to meet our formatting guidelines.

- please upload the manuscript text as an editable doc file
- please upload both your main and supplementary figures as single files; please upload your table files as separate editable doc or excel files, or make sure they're included in the doc file of your manuscript
- please add a separate section with the figure legends to your manuscript
- please add ORCID ID for secondary corresponding author-they should have received instructions on how to do so
- please rename your EV figures as supplementary figures and adjust the figure callouts in the main manuscript text accordingly
- please add a callout for Figure 1A to the main manuscript text
- The reference to Möller-Kerutt, et al. should be removed from the bottom of the current Figure EV3, this is already in the main Reference list
- please remove the "Paper Explained" section

A. FINAL FILES:

B. MANUSCRIPT ORGANIZATION AND FORMATTING:

Sincerely,

December 13, 2022

RE: Life Science Alliance Manuscript #LSA-2022-01649-TRR

Prof. Thomas Weide
University Hospital Münster
Molecular Nephrology
Albert-Schweitzer-Campus 1, A14
Muenster D-48149
Germany

Dear Dr. Weide,

Thank you for submitting your Research Article entitled "Loss-of-surface-transport is a main cellular pathomechanism of CRB2 variants causing podocytopathies". It is a pleasure to let you know that your manuscript is now accepted for publication in Life Science Alliance. Congratulations on this interesting work.

DISTRIBUTION OF MATERIALS:

Again, congratulations on a very nice paper. I hope you found the review process to be constructive and are pleased with how the manuscript was handled editorially. We look forward to future exciting submissions from your lab.

Sincerely,
